# The Datafication of Newsrooms: A Study on Data Journalism Practices in a British Newspaper

Ahmet Buğra Kalender [1,2]

1   School of Arts, English and Languages, Queen's University Belfast, Belfast BT7 1NN, Northern Ireland, UK; abkalender@protonmail.com

2   Institute of Social Sciences, Selcuk University, 42250 Konya, Türkiye

**Abstract:** This study investigates the function of data journalism in a UK newsroom using Bourdieu's field theory. The collection of study data was conducted through in-depth interviews, utilising a qualitative research methodology. The data obtained revealed that data journalism, a sub-field of journalism, continues to develop in an interdisciplinary structure and creates a new type of habitus (data habitus) within the field of journalism. This study also shows that the data journalism team in the newspaper has moved from being niche to being established as one of the most active and effective main sections of the newsroom, and that data-driven journalism has the potential to influence other teams. Lastly, this study suggested that the newsroom is undergoing a process of datafication by indicating the newspaper's intention to develop data skills beyond the data journalism team.

**Keywords:** data journalism; data-driven journalism; ddj; datafication; Bourdieu; data habitus; newsroom; UK





## 1. Introduction

Journalism is currently experiencing a profound transformation driven by digital advancements. This case entails a transformative process that not only impacts the technological instruments employed in the production of news but also the journalists themselves. Nevertheless, our present knowledge of the digitalisation phenomenon remains insufficient, particularly in regard to how journalists interpret and adapt to emerging norms and skills from other disciplines (Lindblom et al. 2022). In recent years, the production and distribution of news have experienced substantial changes as a result of the emergence of digitisation. This has led to the emergence of novel approaches in news production and publication, like data journalism (data-driven journalism). Data journalism is a nascent field that integrates expertise from multiple fields, such as social sciences, data science, data and computer sciences, data analytics, information design, and storytelling (Heravi and Lorenz 2020). One common strategy for making long-term investments in data journalism skill development is the adoption of creative software. At the same time, in order to produce interactive and visual stories, most newsrooms rely, to some extent, on outside platforms (de-Lima-Santos et al. 2021). If there is a problem with the code that is responsible for the visualisation, then the visualisation will either vanish or produce an error. The majority of award-winning data journalism contains dynamic and interactive graphics, which are built on and hosted by a variety of internal and external platforms, according to research conducted on the winners of the data journalism awards (Ojo and Heravi 2018). This relationship not only between newsrooms and other platforms is crucial for data journalism but also for data politics, media ecosystems, and newsroom infrastructure, which become critically important for conducting quality data journalism (Chiumbu and Munoriyarwa 2023). Extensive scholarly investigations have been conducted in the past decade, focusing on data journalism as a prominent emerging practice within the realm of news production. The expeditious assimilation of data journalism within contemporary newsrooms is instigating a novel paradigm shift for conventional news organisations (Kosterich and Weber 2018).

Studies on data journalism can be broadly categorised into two primary directions. Firstly, the emergence of data journalism practices has led to theoretical discussions on the subject in the journalistic field. Researchers in developed countries have focused on what skills are needed to produce data-driven news and the prospects for the future of data journalism in newsrooms. These researchers continued the theoretical debate on the subject and tried to define this new sub-field of journalism in addition to the early practices of data journalism (Gray et al. 2012; Howard 2014; Baack 2015; Coddington 2015; and Lewis and Westlund 2015). The second phase of studies on data journalism is in the form of case studies examining its status in the global north and global south. Although there have been studies examining the practices and status of data journalism in different developed countries, studies conducted in the UK have carved out a significant place in the literature (Knight 2015; Hannaford 2015; Borges-Rey 2016, 2020; Arias-Robles and López 2021; and Tong 2022). The aforementioned studies have undertaken an analysis of data journalism practices in the United Kingdom by gathering data directly from the field. These scholars analysed various aspects, such as the status of data journalism in mainstream and local media, the challenges faced by journalists, and changing journalistic skills. Furthermore, it is worth noting that there exists a limited body of research that explores the realm of data journalism through the use of Bourdieu's field theory (De Vuyst 2018; Fahmy and Attia 2021). In a scholarly study, De Vuyst (2018) conducted an evaluation of the sub-field of data journalism, specifically examining its relationship to gender roles within the framework of Bourdieu's field theory. Similarly, Fahmy and Attia (2021) also used Bourdieu's field theory to analyse experiences encountered by Arab data journalists within the news ecosystem, focusing on the changing of their underlying beliefs and values, known as Doxa.

We do not know exactly why and how data journalists have moved into this field, as well as where the change to data-driven news-producing journalists positions themselves in newsrooms. Furthermore, within a news organisation that has embraced the practice of data journalism, there is a lack of detailed knowledge regarding the specific ways in which journalists influence the functioning of the news operation during this process. Therefore, this study sought to examine the manner in which the data journalism team inside a news organisation based in the United Kingdom has attained a position of utmost significance within the newsroom, employing Bourdieu's field theory as a theoretical framework.

The research questions that were addressed in this study are as follows:

**RQ1:** *How is data journalism integrated into the newsroom?*

**RQ2:** *How are data journalists transforming the news operation of the newsroom?*

## 2. Data Journalism

Data journalism is a journalistic practice that integrates the digital realm of communication with the utilisation of numerical data and visual representations. Scholars have made attempts to establish a definitive definition for this emerging phenomenon in the realm of journalism; nevertheless, a consensus in terms of conceptual understanding has not yet been achieved (Fink and Anderson 2015; De Maeyer et al. 2015; Splendore et al. 2016; and Bradshaw 2017). The widely accepted concepts in the media and journalism literature are data journalism and data-driven journalism (Coddington 2015). Data journalism is a specialised field within journalism that has arisen in response to the growing prevalence of data generation and quantification. In essence, data journalism encompasses the practice of utilising interactive visual storytelling, statistical analysis, 3D maps, and various other methods to convey news and information that is rooted in data (Larrondo-Ureta and Rodríguez 2021). In this way, some professionals who were outside the field of journalism became acquainted with data journalism and started to turn towards this field (Appelgren and Lindén 2020). The main reference to the origin of this concept is seen as computer-assisted reporting (CAR) (Howard 2014; Knight 2015; and Anderson 2018). Scholars in this field have commonly characterised CAR as a form of journalistic inquiry

employed in extensive news endeavours, employing many statistical methodologies in its production (Gynnild 2014). Nevertheless, it is important to note that the aforementioned statement does not provide a comprehensive depiction of contemporary data journalism. Coddington (2018) sought to establish a comprehensive understanding of the quantitative trends in journalism through the application of data-driven journalism in a more expansive manner. He expressed his typology as data journalism, CAR, and computational journalism, creating a more functional definition under the umbrella of data-driven journalism.

Transparency and visualisation are the primary themes that are frequently addressed in the realm of data journalism. Data journalists prioritise the central role of data in news reporting as opposed to considering data as supplementary components in news coverage. Consequently, the majority of data is acquired from publicly available sources, enhancing the readers' ability to access data inside news articles and promoting transparency in news reporting (Coddington 2015). The second concept that data journalists touch upon in the news is visualisation. Successful data visualisation methods transform the data presented in news articles, rendering it more comprehensible for readers who may otherwise find the data perplexing. News obtained from large datasets and presented in a written format may especially make it difficult for the reader to understand (Veglis and Bratsas 2017; Veglis and Maniou 2018; and Lopezosa et al. 2023).

The field of data journalism comprises a relatively new set of abilities that involves accessing, assessing, and visualising digital sources. These skills are becoming increasingly vital in a sector where traditional journalism skills are no longer sufficient. It is not intended to serve as a substitute for news organisations that have already established themselves; rather, it is intended to supplement them. Data journalists have the ability and responsibility to come closer to their sources in this age when many sources are digital (Al Jwaniat et al. 2023). There is the possibility for journalists that the using of data has the capability of transforming an intangible concept into one that can be comprehended and linked to by a large audience. Deconstructing the background of a complicated event, such as a riot or a political discussion, exposing falsehoods, and assisting everyone in identifying viable solutions to complex problems are all things that they are capable of carrying out (Vural and Masip 2021).

The utilisation of data in the generation of news has become increasingly conspicuous in comparison to previous times. Terms such as big data, data collection, data scraping, data storage, data analysis, data management, and data visualisation have become much more familiar to contemporary newsrooms. There has been an emergence of data-related terminology in newsrooms, alongside the introduction of novel positions and job descriptions that were previously non-existent in this domain. News production processes have witnessed the active participation of employees possessing diverse skill sets unrelated to journalism. These individuals have assumed significant roles and contributed to the overall news production endeavour (Tandoc 2019). Some researchers have defined the necessary qualifications as a result of the widespread use of data journalism in newsrooms. Among these, Broussard and Boss (2018) highlighted 'programming languages' like Javascript or Python; Boyles and Meyer (2017) emphasised 'technical skills' such as data visualisation and interface design; Faria Brandão (2019) referred it 'to be data literate'.

One of the initial great expectations for data journalism was that it would be recognised as the journalism of the future (Knight 2015). But it has since been argued that this anticipation was not entirely fulfilled within a certain timeframe, and that instances of data journalism were rarely observed in establishments beyond prominent news organisations (Stalph and Borges-Rey 2018). However, it is plausible to argue that the global occurrence of COVID-19, occurring about one year after the aforementioned period, could have exerted a substantial influence on the field of data journalism on a global scale. Thus, while data-driven journalism continues to spread among mainstream media, it has also been included in the local media ecosystem (Arias-Robles and López 2021; Stalph et al. 2022; and Thäsler-Kordonouri and Barling 2023). In addition, there is a growing prevalence of alternative media (de-Lima-Santos and Mesquita 2023; Wu 2023) and radical media/activism

(Jeppesen 2023; Jeppesen and Sartoretto 2023) that operate beyond the realm of professional journalism, specifically within the domain of data journalism practices; this trend has been observed to be on the rise. Based on these different studies, it can be asserted that data journalism is seeing a growing prevalence on a global scale. Furthermore, the production of news content that is centred around data is witnessing an upward trend not just inside major news organisations but also across diverse newsroom settings.

**3. Theoretical Framework**

Bourdieu's field theory seeks to elucidate the dynamics of power and dominance among agents within the social space, as well as the subsequent reproduction of social relations that arise from these interactions. According to this theory, fields in the social space (field of journalism), the dynamics of social forces (newsrooms as a site of struggle), and agents (journalists) that improve in the social space coexist (Bourdieu 1977). Bourdieu (1986) posited that agents, through the utilisation of their many forms of capital (economic, social, cultural, and symbolic), mediated by their habitus, become entangled in relationships of dominance within a given field. The accumulation of capital within this field empowers agents to engage in competitive interactions with other agents within the same field. Bourdieu (1998) likened the fields within the social space to a kind of game. Players' strategies use their habitus and capital and thus gain power to compete within this field. From another point of view, players have to obey the rules of the game in order to exist in this field. The rules of this game may change over time, but this concerns the player, not the game. The player adapts and repositions on the field according to the changing rules. As in every field, journalism has its own internal dynamics, and Bourdieu (1999) saw this field as a "*microcosm with its own laws, defined by its own position in the world at large and by the attractions and repulsions to which it is subject from other such microcosms*" (Bourdieu 1999, p. 39, as cited in Tandoc and Oh 2017, p. 2).

Bourdieu's field theory provides a significant framework for the scholarly examination of journalism, which can be defined as a social space within the field of study (Benson and Neveu 2005). Journalism can be analysed just like any other social field. Actors in the field (journalists) can compete with other actors in the field and achieve a higher position by accumulating different types of capital and changing their habitus (De Vuyst 2018). In other words, to exist in a field means to reshape yourself (Bourdieu 2005). Habitus refers to the cognitive abilities, beliefs, and dispositions of the agent acquired in a social space. Its past experiences in the recesses of the agent's mind are shaped as a matrix of actions that influence their future life and choices. So, habitus is a holistic concept. Agents' experiences and behaviours in daily life, together with their past experiences, constitute their habitus (Obembe 2013). Although the agents' prior experiences and cultural capital shape their habitus, it is not unchangeable. New entrants into this field, together with interactions from other fields, can slowly change the existing habitus (Webb et al. 2002).

In the field of journalism, habitus can generally be expressed as the internalisation of news value and mastering the news game. This habitus is often shaped in newsrooms and places where journalism training is provided (Schultz 2007 and Hovden et al. 2016). In recent years, the practice of journalism has undergone significant changes due to advancements in technology, resulting in a transformative effect on the journalistic habitus. Thus, these changes started to find their place both in newsrooms and in the journalism curricula of universities trying to keep up with the media ecosystem (Lindblom et al. 2022). In other words, habitus is a structure that shapes an agent's perspective on changes occurring in a field and is in constant interaction with various forms of capital. Bourdieu (1984) also maintained that different types of habitus can emerge depending on the conditions of formation within the field and referred to this as a 'secondary habitus' or 'habitus 2'. From this point of view, it can be stated that journalists have a sub-habitus in the field of journalism just as they have a general habitus as agents (Marchetti 2005; Schultz 2007; Willig 2013; and Kumar and Haneef 2018). Therefore, it is possible to think that there will be specific journalistic habitus, such as an 'editorial habitus', a 'reporter habitus', or an

'intern habitus' (Schultz 2007). Sub-habitus types within the field of journalism have been analysed by some researchers. Some of them are video habitus (Bock 2012), online news habitus (Møller Hartley 2013), mojo habitus (Kumar and Haneef 2018), and DSLR habitus (Vasudevan 2019). The different forms of habitus here not only explain the general perspectives, skills, and behaviour of journalists with regard to news reporting but also characterise a holistic understanding, revealing the contradictions, challenges, and competition in this sub-journalistic field (Willig 2013 and Kumar 2018).

## 4. Methods

This study focused on the data journalism team of an internationally recognised newspaper situated in London. The primary objectives were to gain insights into the extent of journalists' involvement in data journalism and to ascertain the role and significance of data journalism inside the newsroom. Therefore, the data for this research were obtained using the qualitative research method through the interview technique. Within this framework, interviews were conducted using semi-structured questions that allowed participants the freedom to express their views and experiences, and the obtained data were analysed using the descriptive analysis technique. The semi-structured interview questions used in this research emerged as a result of the literature review. Participants for the interviews in this research were selected through purposive sampling. Purposive sampling is one of the most commonly used sampling methods in qualitative research, allowing for the selection of individuals that are most suitable for the research purpose (Guarte and Barrios 2006; Campbell et al. 2020). In total, 13 journalists were interviewed, and all of them were selected from various members of the 'Visual and Data Journalism Team' (e.g., editor, senior data journalism engineer, data journalist, data visualisation journalist, data scientist, graphics journalist, and graphic artist). In order to ensure confidentiality, several measures were implemented due to the fact that all participants were members of a cohesive team consisting of around 30 individuals operating within the same news organisation. All participants were coded as journalists (Js) in order to prevent the possibility of revealing their identities in case the participants were coded with their job descriptions. In addition, the name of the newspaper that constitutes the study group was kept confidential in order not to disclose the identities of the participants. All the interviews were conducted over a 3-month period between May and July 2023. A total of 12 interviews were conducted in person, but a journalist, J7, was interviewed both in person and online. The duration of the interviews typically ranged between 30 and 60 min, during which audio recordings were made. Subsequently, all the audio recordings were transcribed. As one of the participants, journalist 4 (J4), was unable to spare time for their interview, the questions were sent via email, and responses were received in this way. In the process of coding the data, a manual thematic coding was made by considering the themes created before the interviews. Then, thematic coding was made by paying attention to the common points between the concepts, and the themes were thus determined. Direct quotations were also used to reveal the participants' original thoughts on the subject. I like some researchers (Kumar and Haneef 2018; Topić and Bruegmann 2021), resorted to thematic analysis to categorise several concepts, such as field, capital, and habitus. This allowed the observation of the newsroom's data integration process under various headings. A dataset can be analysed with the help of thematic coding, which is beneficial for discovering essential aspects, contrasts, and similarities (Boeije 2010; Braun and Clarke 2006). Thematic codes should be created beforehand according to the themes examined in the study (Hujanen et al. 2022). The data collected as a result of the interviews with the team members were coded thematically, and a meaningful path was drawn for the data journalism sub-field.

Heravi and Lorenz (2020) tried to understand why well-known news organisations in the world are more frequently preferred by researchers to collect data in studies on data journalism. These authors suggested that a primary factor contributing to the preference for large news organisations in data journalism research is likely their ability to generate a greater quantity of data news content focused on higher volumes. These researchers

also suspected that this may be due to a positive perception of organisations with higher reputations in the news media. However, these researchers have indicated that they currently lack any substantiated evidence or sources to support their claims. There are three main reasons why I chose this news organisation for this study. My first and second reasons coincide with the reasons put forward by Heravi and Lorenz (2020). My other reason is that, according to the field theory, powerful structures in the media not only shape public opinion but can also play a decisive role for other media organisations (Bourdieu 1999). The visual and data journalism team of the newspaper was preferred as the study group in this research, as it brings together a team from many different disciplines, produces practices that address many aspects of data journalism, and has the potential to influence other institutions and organisations in the field of data journalism.

## 5. Analysis

### 5.1. Transition to Data Journalism

The advent of technology has significantly broadened the scope of news production, leading to notable advancements in the area of journalism. Data journalism started to function as a separate department in newsrooms in 2010 and onwards, and it continues to be increasingly influential. On the other hand, economy/business-orientated newspapers have been associated with data and statistics for a very long time (Arrese 2022; Munoriyarwa 2022). The newspaper under analysis originated as an economy/business publication, establishing a historical connection between the newspaper and data dating back to its inception. However, the newspaper's data journalism team in the modern sense was formed in 2019 in an interdisciplinary manner with the merger of many sub-units. This team was created by merging many different teams that previously existed within the organisation (print graphics, online graphics, interactive news, and statistics) into a single team (J7 and J10).

A considerable number of journalists, who initially embarked on their journalism careers without specialising in data journalism, have subsequently transitioned to this particular domain (Morini 2023). The majority of journalists within the visual and data team (J2, J3, J4, J5, J7, J8, J9, J10, J12, and J13) indicated that they did not initially embark on their professional careers as data journalists. However, these journalists, who have been introduced to data-driven journalism in various ways, do not have very different views about their transition to this field. The participants generally think that data journalism makes the news more 'provable' (J1 and J8), 'credible' (J6, J11, and J13), and 'creative' (J9 and J12). Therefore, the idea of increasing the credibility of the news and presenting the news in a more creative way has been the main reason for them to turn to data journalism:

> "I love the way you can use data to interrogate a hypothesis and discover answers to questions, then take that same data and visually present your findings to readers". (J2)

> "I really like the data. Because I have just always liked it. Feels more honest than individual stories". (J11)

> "When I started in newspapers as a graphic artist, I was never really interested in data at all. But when I joined this team, I realised the possibilities of what we could do with it and how creative we could be. Now, literally I treat data as a completely separate art form, truly seeing it as a form of artistic expression". (J12)

Previously, it was quite ordinary to see journalists working in different positions and roles within the field of journalism. However, the advent of digital advancements in this domain has transformed the aforementioned mediocrity due to the involvement of numerous individuals from diverse backgrounds beyond the realm of journalism in news production. Data-driven developments in the journalism industry not only enable journalists to integrate into this field but also transform people who did not start their career as journalists into data journalists. Participants who previously worked outside the field of journalism approached this issue from a similar perspective. J8, who started

their professional life in the field of public policy and worked as a data analyst in public institutions, said that he has always supported the idea of public debates using 'proof', which is why he turned to data journalism. Moreover, they believe that data-driven testing of claims made by individuals or organisations is a more 'honest' journalistic practice:

"I kind of see data journalism as a way of keeping journalism honest". (J8)

Another journalist, J13, started his career in computer science and said that he had no interest in journalism at the time. Approximately six years ago, the journalist in question embarked upon an examination of data journalism practices, subsequently discovering an interest in this domain and then transitioning their focus to this particular subject. Another participant, J4, who started his working life in the field of computer science, had a similar experience:

"I had a previous career as a software developer... My role had nothing to do with journalism in fact, I was about as far away as you could get, working on their content API. However... ended up chatting with some journalists there and discovered something called data journalism, which combined something I knew about with something I cared about".

*5.2. The Forms of Capital for Data Journalists: Cultural, Social, and Symbolic*

The field theory conceptualises society as a dynamic field whereby many struggles and conflicts take place. In order to develop a strong position within a certain field, an individual (agent) must effectively utilise several forms of capital, including economic, cultural, social, and symbolic capital (Bourdieu 1986, 1996). These different forms of capital gain meaning in determining the position of agents in a social space. For Bourdieu (1998), economic and cultural forms of capital are more dominant than others. To exist in the field of journalism, the agent (journalist) must possess the necessary knowledge and journalistic skills (cultural capital). Furthermore, apart from possessing cultural capital, journalists also require social and symbolic capital to acquire status and recognition, thereby distinguishing themselves within their profession. Therefore, several types of capital are important for the positioning of the journalist in this field (Maares et al. 2021). Although having the necessary types of capital to exist in this field is one of the determining factors, this alone is not enough. This is why Bourdieu (1996) has often associated capital and capital types with habitus. The type and amount of capital obtained by the journalist or the total distribution of capital within the field is the most important factor in the formation of habitus (Schultz 2007).

Cultural capital symbolises the education, achievements, and skills of the agent from the past. The newspaper's visual and data journalism team is made up of journalists with different skills, and most of the team members have a bachelor's or master's degree in journalism. There are also journalists with degrees in computer science, statistics, geography, economics, etc.

Undoubtedly, it is necessary to have a number of skills to perform well in data journalism. Previous studies (Veglis and Bratsas 2017; Heravi 2019; Heravi and Lorenz 2020; and Cardoso 2022) have shown that data journalists or aspiring data journalists should have multiple skills, if not all the skills in this field. All the journalists in the newspaper's visual and data team support this view. One of the participants, J7, stated that they have different expectations from team members according to their positions, but the common expectation for team members is that they should have the ability to process data in a spreadsheet. Also, J10 argued that coding and visualisation are a unique form of storytelling in data-driven news production and that the data journalist with these skills will be different from the others. Becoming an expert in one of the skills needed to carry out data journalism provides a significant advantage for journalists in this process:

"My advice would be to pick a skill and focus on getting really good at it, rather than trying to learn how to handle data, build graphics, code, illustrate etc. It makes the journey into data journalism more manageable". (J2)

The fact that journalists working in news organisations receive professional training and develop their skills in the areas they need is a situation that can be identified with cultural capital. Producing news using technological tools also implies the necessity for journalists to receive the appropriate training (software, coding, data analysis, visualisation, etc.). Journalists working on the team said they receive in-house training on the new tools, programmes, and skills they need (J3, J6, J8, and J10). In addition, if journalists want to attend a conference or a training course that is useful as a career development opportunity, the newspaper finances this from its training budget. Participation in these activities is encouraged by the organisation even if it is not compulsory (J7, J8, and J12).

Social media platforms are a very important area for both news organisations and journalists. Similarly, it is noteworthy that X/Twitter has emerged as a significant platform for data journalists to obtain news information (Gehrke and Benetti 2020). Of note, X/Twitter has a critical importance for journalists' social capital accumulation (Barnidge et al. 2020 and Maares et al. 2021). The team has an account on X/Twitter separate to the newspaper's main profile where they publish data-driven news, and almost all of the team's journalists have personal profiles that they use actively. Journalists' profiles on social media not only contribute to their own social capital but also to their collective social capital. Collective social capital increases news visibility and builds digital audiences. Therefore, social media assumes a significant function in facilitating engagement with readers (Kumar and Haneef 2018):

> "Direct interaction with readers is limited largely to social media (particularly X/Twitter) and comments left on articles which we are encouraged to review and engage with if they raise specific points and occasional direct email correspondence. These may include claims of error that must be checked, corrected if necessary, and responded to in all circumstances". (J7)

Moreover, some of the team's journalists are supporting the 'journocoders' event and taking on important roles in the organisation (J4 and J5). The journocoders event, which is held once a month in London, serves as a kind of workshop for data journalists from different institutions and journalists or candidates who want to learn data-driven journalism. Journocoders organise a monthly event announcement on Meetup, where participants can register for the free event. The main aim of this event is to teach the participants how data can be used in news production through practical examples. While this is important for the individual social capital of J4 and J5, it also contributes to the collective social capital of the team.

Another type of capital for data journalists is symbolic capital. Economy/business-based newspapers have always been in contact with data from the past to the present (Arrese 2022). This may be due to the historical background that led the newspaper in this study to a remarkable position in data journalism. J7 argued that many of the techniques of data journalism in the modern sense were developed by this newspaper:

> "These older teams had much longer histories. The graphics and statistics or data functions have existed at our newspaper for many decades, while the online graphics function has existed since the website was founded in the mid-1990s and the interactive team was created around 2006. Many of the computational techniques today called 'data journalism' were first adopted at our newspaper by these teams".

Furthermore, the awarding of journalists, either individually or as a team, contributes to the accumulation of symbolic capital, such as 'prestige' and 'status', in this field (Siapera and Spyridou 2012). This team has won some awards in recent years (*Digital Initiative of the Year, Data Journalist of the Year, Specialist Journalist of the Year, Wincott Awards, and Royal Statistical Society Awards et al.*). This contributes both to the newspaper's prominence in the field of data journalism and to the influence of the journalists working there. Becoming recognised and rewarded by fellow journalists allows them to rise to better positions in their field (Singer 2008; Kumar and Haneef 2018).

*5.3. The News Operation of the Visual and Data Journalism Team*

The visual and data team consists of more than 30 journalists from different disciplines and is divided into several sub-teams of different sizes. Their sub-teams are listed as data reporting, data science, graphics, user experience (UX)/user interface (UI), software development, and visual storytelling. Among these team members are journalists with different titles who have various competences. New job descriptions added to the digitalised news ecosystem have become increasingly diverse with data journalism practices. In addition to the traditional employment roles of editor, data journalist, visual journalist, and graphic designer, the team has new additional positions, including senior data journalism engineer, editorial data scientist, and newsroom developer. Although the newspaper is a London-based organisation, it includes team members from different countries. All the team members meet once a week in one meeting and are accompanied online by team members from Hong Kong and New York. Additionally, there is a daily meeting in London every morning, and senior members of the team attend this meeting. This meeting is recorded so that the New York team can watch it later on the day when they start work. Additionally, the various specialised sub-teams can organise smaller, more focused meetings throughout the week, and the team's managers can also regularly participate in the planning meetings of other teams in the newsroom. These teams are made up of a combination of reporters, cartographers, illustrators, designers, and developers who work with colleagues across the newsrooms using graphics and data to discover, investigate, and explain stories. This can be anything from quick turnaround graphics for breaking news stories to in-depth visual investigations that take months. Each team has a specific focus, but there are a number of skills (data handling and visualisation, for example) that everyone has (J2, J7, and J9). There are no restrictions on the types of stories that the team can cover. The team can include any type of news (politics, business, health, environment, science, sports, etc.) that fits the newspaper's concept of newsworthiness and requires data analysis or data visualisation (J11 and J13).

The acquisition of data journalism skills is considered significant by certain journalists since it enables them to adapt to the dynamic news industry and enhance their professional standing within newsrooms. Professionals who started their careers in different departments of journalism thought that by moving to this field, their data-driven journalism skills would make them more prominent in newsrooms. Some journalists (J1, J3, J5, and J10) think that they are able to use data to create more 'original', 'sophisticated', and 'innovative' stories that other journalists cannot discover. The fact that data-driven reporting is a less common skill in the news media makes these journalists think that they are different in newsrooms:

> "I was discovering things that other journalists couldn't find using data. Since it wasn't something many people were doing in the media world, I felt that focusing on it was a smarter choice and then it made sense to continue in this field". (J3)

> "Besides writing news, I wanted to combine my computer skills with it. Computer skills are less common in the journalism industry compared to writing skills. This makes you stand out". (J5)

Data-driven journalism undoubtedly requires a lot of software. The team benefits from many computer programmes, and journalists use them according to their areas of expertise. The aforementioned software encompass widely recognised applications like QGIS, Adobe, Blender, Flourish, Datawrapper, and R. Journalists make use of them according to their areas of expertise. In-house trainings are, similarly, organised for data journalists to learn the newly integrated programmes in the newsroom and to develop software (J5, J8, and J12).

News organisations publish their editorial policies transparently and commit their employees to abide by them. In addition, there may be some guidelines that journalists must follow specific to the departments in which they work. There exist fundamental principles that govern the team, particularly those that establish the parameters of data visualisation. The team possesses a comprehensive collection of stylistic guidelines for charts and maps,

which has been meticulously cultivated by team members over an extended period of time. This emphasises significant factors that those working with data, particularly those employing data visualisation, should take into account (J6 and J11). Moreover, most of the graphics and visualisation tools available to employees work in a way that makes it almost impossible to create graphics that do not fit the corporate brand guidelines. Authentic data visualisation at the newspaper helps to produce news stories faster. J7 explained the situation as follows:

> "News organisations' own efforts at templating and the growing availability of off-the-shelf data visualisation technologies have enabled much faster production of complex visual forms that would, in the past, have required bespoke coding. We have done both these things. Some years ago, we built a large library of code for various types of data visualisations. We also built a code template for any custom-designed visual stories. Both of these dramatically accelerated our production capability compared to the previous practice of bespoke production for each new item".

Every discipline possesses its own unique dynamics of challenges. Simultaneously, it can be argued that some complex dynamics pertaining to data journalism influence the functioning of news operations. One of the primary challenges in the field of data journalism pertains to the absence of well-defined boundaries delineating areas of expertise. While journalists often excel in data-driven tasks, there are instances where they may require precise details and comprehensive analyses (J2, J3, J7, J9, J10, and J11). Most of the data journalists at the newspaper have the ability to produce data-driven news, but the fact that journalism consists of many different types of news often creates the need for specialisation within this field. For this reason, team members produce news in collaboration rather than individually:

> "I think one of the big challenges is having a medium as our specialism, rather than a beat. It means that we tend to be working on the biggest stories of the moment, but it's harder to think of ideas and exclusive angles when you're regularly changing topics. Luckily the data often gives you a unique way in". (J2)

> "One of the challenges is that you're not a specialist in a topic, you're a specialist in a skill -- so you don't have the depth of knowledge, or the contacts, that someone working a specific beat does. This is why most of my stories are collaborations with other reporters that can bring that side of things". (J3)

> "Most stories at the newspaper are collaborations between journalists with different specialists knowledge... the same is true of any story involving data or graphics: the technical specialist journalists in the visual and data team work works with a subject-matter specialist from another team". (J6)

Data-driven news production plays an important role in the newspaper's general journalistic approach. The newspaper allocates a significant budget and facilities for the visual and data team. This team has become one of the largest and most influential teams in the newsroom and has the power to influence other teams. Therefore, it is no longer possible to say that data journalism is a niche practice for the newspaper. In fact, some journalists believe that data journalism is no longer niche:

> "I don't think data journalism could be considered a niche field anymore. We are one of the largest desks in the newsroom and our work shapes our newspaper coverage every single day". (J2)

> "There was a time, not so long ago, when data journalism was new and exciting. It's not a new thing anymore; it has matured into a standard, everyday part of a modern newsroom". (J4)

This newspaper uses many computer programmes, software, and services to produce news. However, the visual and data team has a different function compared to other teams

in the newsroom. Apart from using the many tools necessary to produce data-orientated news, the team shows another example of being 'innovative' and 'creative' in the newsroom by writing original codes:

> "There are many sorts of bits of software that we use, but I think the crucial thing that sets our team apart from others in the newsroom is the fact that we're also writing software. Rather than just using other people's software where we're writing code in JavaScript than our in particular and using that to do the fundamental things of finding, analysing, and telling stories with data". (J8)

On the other hand, COVID-19 has severely affected journalism, as well as other sectors (Bernadas and Ilagan 2020; Perreault and Perreault 2021; García-Avilés 2021; Perreault et al. 2022; and Trifonova Price and Antonova 2022). The pandemic process seems to have created a serious breakthrough, especially in the context of data journalism (Westlund and Hermida 2021; Pentzold et al. 2021; and Bisiani et al. 2023). Nevertheless, the hybrid working model seen in the COVID-19 modus operandi and its aftermath is not a completely new model for the newspaper. Working in collaboration with centres outside of London, the team had experience working with the hybrid or remote working models. Although members of the visual and data team were accustomed to this working model before COVID-19, after the pandemic, they have been at the newspaper building on average three days a week (J1, J6, J7, and J11). This provides further evidence that, as Hendrickx and Picone (2022) suggest, a hybrid working model may become permanently operational for newsrooms post-pandemic:

> "Yes, we already had remote working processes before the pandemic. However, many of the processes we currently have really emerged during the pandemic". (J5)

According to these journalists, although this process brought some changes, the data team seems to be one of the teams that adapted to the hybrid working model the fastest:

> "The team had already worked in a highly distributed way, with members working in New York, Hong Kong, Manila and Madrid, so it had been an early adopter of tools like Slack and Github designed for the collaboration of distributed teams. The shift to working from home and hybrid working was therefore not difficult—perhaps easier than for some other teams more accustomed to largely in-person communication". (J7)

The pandemic has resulted in a heightened need for data-driven news and a surge in interest in data visualisation, significantly driving the field of data journalism (Liu 2021). These participants hold the belief that data journalism has effectively addressed a significant global demand during the pandemic. Furthermore, it is believed that the proliferation of data journalism on a global scale will be facilitated by the emergence of effective data visualisation:

> "The pandemic brought data journalism to the forefront and news organisations with strong data and visual teams really succeeded during the pandemic". (J3)

> "Data news during the pandemic became one of the biggest stories in the world. Perhaps it was the biggest news of our lives. This greatly demonstrated the importance of data, good data journalists and visual journalists". (J8)

Additionally, the visual and data team has been the team that has played a significantly active role in the pandemic within the newspaper, and this has brought the team to a prominent place within their newspaper. Moreover, all of the participants think that data journalists will have an even more effective position in the newsroom in the future. Here, factors such as 'digitalisation' (J1), 'visualisation' (J12), and 'quantification' (J9) create the need to produce more and more data-driven news in the newsroom:

"I think the COVID-19 process elevated our status within the newspaper as a team. On the other hand, I believe our status as a specialisation in journalism also increased". (J6)

"I think there are more people interested in entering this field, and it will be interesting to see how it develops further. Combining data journalism with other newsrooms is something that needs to happen, and I expect to see experts in data and visuals in other newsrooms in the next five to ten years". (J10)

"Not only our newspaper but even newsrooms with limited resources can now undertake large visual and data-driven projects. So, local newsrooms have become a really interesting field for data journalism". (J5)

"Data journalism will be a more integrated process for newsrooms, and I believe we'll be able to sit down with the world desk and immediately collaborate on their fantastic ideas. The combination of these thoughts feels like a natural progression, so we will be fully integrated". (J8)

Lastly, in order to encourage all journalists within the newspaper to develop certain data-driven reporting skills, the newspaper organises various training sessions in the newsroom that are open to all journalists and implements certain protocols:

"We have an extensive editorial training programme open to everyone in the newsroom. This includes voluntary training on 'newsroom numeracy', spreadsheet skills and use of our internal chart-making tool. In addition, all of our graduate trainees and some other early-career journalists receive a one-week 'data boot camp' as part of the training provided". (J7)

## 6. Discussion and Conclusions

The field of journalism, like a game, is an area of struggle for various actors. In order to get ahead in the game, the parties play on their existing gains. If these gains can propel them to a more advanced stage in the game, it is linked to the strength of their capital and the structure of their habitus (Bourdieu 2005). This study utilises Bourdieu's field theory to examine data journalism and the positioning of journalists in the newsroom. In order to conduct this, the main motivations for journalists working in a UK-based news organisation to switch to data journalism were examined, and it was found that the capital and types of capital that journalists possess have an important place in the transitions between journalistic fields. This research further elucidates the transformation of news operations following the incorporation of data journalism practices into the newsroom.

### 6.1. Transition to Data Journalism and Accumulation of Capital

To address RQ1, the initial focus was on exploring the ways and reasons journalists embraced data journalism, albeit through diverse views. For Bourdieu (1998), the competitive environment in the field of journalism, as in every field, requires journalists to accumulate capital in order to ensure that they are positioned according to the new conditions in their field. This study shows that with the institutionalisation of data journalism from newsrooms in the changing field of journalism, journalists add data-driven capitals to their existing capital accumulation and display them in their field. The data journalists employed by the newspaper demonstrated an awareness of the evolving demands within their profession and proactively acquired the necessary competencies. Alongside this, the data-driven journalism process in the current newspaper has added new roles and responsibilities to the field and has endeavoured to keep its institutionalisation in news production at a high level by providing regular in-house training to its employees. Even though the newspaper has many of the computer programmes and software used for data-driven journalism, it is looking for ways to be more efficient, fast, and creative in the field by developing its own software. Although the common expectation for each member of the team is to have potential data processing skills, most of the employees have different

skills and even consist of professionals from different continents. This enables the interdisciplinary team to conduct news operations, often in collaboration, and it is demonstrated to closely parallel some studies on the outcomes of integrating digital and data journalism into newsrooms (Primo and Zago 2015; Arias-Robles and Carvajal 2023). This study shows that data journalism practices have begun to spread throughout the newsroom, albeit to a certain extent, in a British newspaper examined using Bourdieu's field theory. Therefore, it can be said that for most journalists in the newsroom of this newspaper, being able to report data-driven news or at least to contribute to this process has become a necessity. However, this does not indicate that data surpasses journalism. According to Heravi (2017), journalism and research are the two main skills that any aspiring data journalist should possess. The most crucial aspect to be developed thereafter is familiarity with data and the ability to create news using data.

*6.2. Transforming News Operation*

The most important detail for RQ2 is that data journalists do not represent a minority group in the newsroom. This study shows that, in alignment with de-Lima-Santos (2023), data journalism activity in corporate newsrooms is now more than a niche area. The majority of the participants (J1, J2, J4, J5, J6, J7, J8, J10, J11, and J12) think that data journalism will be integrated into the newsroom's general day by day. As J2 and J4 stated, the team has reached a position of influence that can shape the overall news operation of the newspaper. Although data journalism does not represent a niche field for the newspaper, it may be necessary to have a niche in data journalism skills to stand out in this field. Despite data journalists achieving a pivotal role in the newsroom's transformation, the continuous evolution of the news ecosystem towards a data-centric approach increasingly necessitates the cultivation of innovative and distinct skills based on data. Moreover, the newspaper organises training sessions for all of its employees to improve their skills in working with data. This is one of the most important pieces of evidence that the newspaper wants to fully integrate data into the newsroom. Although this situation is considered a necessity for all data journalists, it not only concerns them but also almost the entire newsroom. All this shows that data journalism is becoming one of the central parts of the newsroom and has the power to seriously influence other teams. It also shows that data journalists are quick to adapt to changing rules of the game (like COVID-19) and, moreover, that they have the potential to influence other departments in the newsroom.

*6.3. Data Habitus: Change and Adaptation in the Newsroom*

Bourdieu (1984) explained habitus as a concept that changes depending on field conditions and then shapes individuals' perceptions and practices. Habitus is a powerful tool that individuals can use to link practices and structural processes. On the other hand, habitus functions as a bridge between the old and the new in the transition between sub-fields, and there are multiple forms of habitus across multiple sub-fields (Marchetti 2005; Schultz 2007; and Willig 2013). Furthermore, habitus can differentiate in the new order (Wu et al. 2019). This study proposes the concept of 'data habitus' on behalf of the secondary/sub-journalistic habitus as a result of the data obtained from the newsroom regarding the data journalism sub-field of journalism. This concept has been used to determine the integration of the concept in newsrooms in the last 15 years, when data started to take place in the journalism industry in the modern sense, to show that data-orientated news production has been institutionalised and to characterise the data-orientated processes added to news production. Additionally, this concept is also important for describing journalists' skills in data-driven journalism (data scraping, data extraction, data analysis, data visualisation, etc.) and their perceptions of their roles in their field. Data habitus shapes both the capital types and the process of journalists' transition to the field of data journalism and also enables this field to become more interdisciplinary.

While the data habitus is effective in addressing expanding market demands and rapidly rising innovative processes in journalism, it also shows that there is room for

improvement for journalists with data journalism skills. The fact that the majority of data journalists specialise in a skill rather than a type of news, while necessitating collaborative news production, also shows that there is a need for data journalists who specialise in a type of news.

All in all, this study has several limitations. Although this study collected data from the employees of one of the more important organisations in the field of data journalism, it only evaluated the process through a limited participant experience. Incorporating Bourdieu's field theory in this area of study may prove to be a productive endeavour for future studies, namely those that partake in further qualitative research methods. In particular, it is important to collect data from a diverse study group in order to make a judgement on the extent to which data-driven news production shapes journalists and newsrooms.

**Funding:** This research was funded by [The Scientific and Technological Research Council of Türkiye (TUBITAK) 2214-A/International Research Fellowship Programme for PhD Students] grant number [1059B142200562].

**Institutional Review Board Statement:** The study was conducted in accordance with the Declaration of Helsinki, and approved by the Ethics Committee of Faculty of Communication, Selcuk University (protocol code 256018/16 March 2022).

**Informed Consent Statement:** Informed consent was obtained from all subjects involved in the study.

**Data Availability Statement:** Data are contained within the article.

**Acknowledgments:** This article is a part of a PhD research thesis that the author is currently undertaking at the Department of Radio, Television and Cinema, in the Institute of Social Sciences at Selcuk University, Türkiye. The author extends their sincere gratitude to Queen's University Belfast, the host institute where they conducted this research as a visiting scholar, and namely, Sue-Ann Harding, the host supervisor. This work has been published through the generous support of MDPI to whom the author extends their sincere gratitude and appreciation.

**Conflicts of Interest:** The author declares no conflicts of interest. The funders had no role in the design of the study; in the collection, analyses, or interpretation of data; in the writing of the manuscript; or in the decision to publish the results.

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
