# Peer review of "The Datafication of Newsrooms: A Study on Data Journalism Practices in a British Newspaper"

_journalmedia, doi:10.3390/journalmedia5010004_

Round 1

Reviewer 1 Report

Comments and Suggestions for Authors

The article addresses a question of evident relevance and provides substantial findings on a field of study such as data journalism. The authors present their ideas in a clear and concise manner, effectively conveying their message to their target audience. The paper reflects the authors' dedication and hard work, as it is evident that they conducted thorough and comprehensive field research to ensure that their findings were accurate and reliable.

However, the following recommendations are made:

Introduction

An excessive overview of the technological evolution of journalism is carried out. It would be more effective to start directly with data journalism. This would allow for more space to be dedicated to presenting the foundations of data journalism in the introduction (concept, history...) and to delve into other aspects such as its integration in newsrooms in the literature review.

Literature review

In line with the above, I think it would be necessary to expand the collection of the main findings from the literature on data journalism. There are various publications that provide an overview of the profile of data journalism professionals, addressing the technological resources they use or explaining in detail how they work within newsrooms. It would be convenient to address in the literature review the subtitles that are later expanded in the results.

Methodology

I understand the interest in preserving the anonymity of professionals, but I believe it would be necessary to at least list the media outlets where the interviewees work. It would also be important to specify whether the interviews were conducted in person or online. And it would be helpful to provide details about which interview (JX) was conducted via email.

Discussion and conclusion

It is necessary to greatly expand the contrast of the results obtained in this research with other previous ones on the same subject, in order to reinforce the coincidences and highlight the contradictions. We cannot only reference theoretical and older works such as those of Bourdieu or Schultz.

Comments on the Quality of English Language

No errors in writing are detected

Author Response

The article addresses a question of evident relevance and provides substantial findings on a field of study such as data journalism. The authors present their ideas in a clear and concise manner, effectively conveying their message to their target audience. The paper reflects the authors' dedication and hard work, as it is evident that they conducted thorough and comprehensive field research to ensure that their findings were accurate and reliable.

First of all, thank you very much for your comments on my work. I have edited my article taking into account all your criticisms. I hope I have made all the changes you requested in the best way.

However, the following recommendations are made:

  • Introduction

An excessive overview of the technological evolution of journalism is carried out. It would be more effective to start directly with data journalism. This would allow for more space to be dedicated to presenting the foundations of data journalism in the introduction (concept, history...) and to delve into other aspects such as its integration in newsrooms in the literature review.

I removed the parts where I drew a general framework about the technological development process of journalism from the article and tried to describe the process directly through data journalism. Thus, I made a more effective introduction to this concept, which I will extend in the literature review.

  • Literature review

In line with the above, I think it would be necessary to expand the collection of the main findings from the literature on data journalism. There are various publications that provide an overview of the profile of data journalism professionals, addressing the technological resources they use or explaining in detail how they work within newsrooms. It would be convenient to address in the literature review the subtitles that are later expanded in the results.

In this section, I have included the outputs of several publications as you suggested, so as to provide a clearer picture of the expectations for data journalism, the key requirements for news production and how data journalists work in newsrooms.

  • Methodology

I understand the interest in preserving the anonymity of professionals, but I believe it would be necessary to at least list the media outlets where the interviewees work. It would also be important to specify whether the interviews were conducted in person or online. And it would be helpful to provide details about which interview (JX) was conducted via email.

I did this research in only one newspaper, which means that all the journalists I collected data from work in the same organisation and in the same department (Visual and Data Journalism Team). The team editor did not want the name of the newspaper to be disclosed and warned me about it several times. Therefore, I have to fully comply with the confidentiality conditions. I have also made additions to the details you suggested regarding the interviews (in person / online / email).

  • Discussion and conclusion

It is necessary to greatly expand the contrast of the results obtained in this research with other previous ones on the same subject, in order to reinforce the coincidences and highlight the contradictions. We cannot only reference theoretical and older works such as those of Bourdieu or Schultz.

The fact that the study was shaped by Bourdieu's field theory caused me to act with reference to Bourdieu in the discussion and conclusion of the concept I proposed. However, with your recommendations, I made references to other recent studies.

Reviewer 2 Report

Comments and Suggestions for Authors

This paper provides a very interesting case study into the penetration and institutionalisation of data journalism in newsrooms, purportedly through the lens of field theory. The interview data are rich in perspectives and will contribute to the growing body of knowledge about the increasingly central status, roles and functions of data journalism in newswork. The conclusion is sound based on the basis of the empirical data.  

That said, I do think the paper still has important work to do:

1. The most important issue is the lack of a coherence between field theory concepts, the RQs and the data analysis (the Findings section). The Theoretical Framework section introduces the key concepts of field theory but offers very little to link readers to the two RQs espoused earlier.

How do the concepts of cultural, social and symbolic capital and their relationship with habitus help to understand the way data journalism is integrated in news processes and practices (RQ1) and is positioned in newsrooms (RQ2). The Findings section makes this a little more confusing, as it is not organised around the RQs, making it hard to follow your narrative. Of course, I can have a vague idea of how they are linked (esp. after finishing reading the conclusion) but that is not enough. You will need to foreground those relationships explicitly in your literature review and then organise your Findings section accordingly.

One way that helps not only to create a more coherent and focused flow but also improve the theoretical contribution of the paper is to (1) delete the two current RQs from the introduction, then (2), as you go through the theoretical framework, pose a set of new, smaller questions according to field theory's key elements; and then (3) structure your analysis of findings according to each of these smaller questions. 

2. The Method (not Methodology, but the way) section needs to provide far more details on the interview procedures and processes. Did you, for instance, use a semi-structured interview schedule? If yes, how were the questions ordered to maximise your ability to elicit thoughts and information from your interviewees? Also, one single generic sentence about the data analysis method is not acceptable. Please flesh out more information about the process of thematic coding and how it was used to draw a meaningful path to field theory. 

3. The paper needs some major sub-editing and proofreading (see below). 

Comments on the Quality of English Language

First, it is too wordy and the pace is at times dragging. There are too many instances in which the same words/phrases are repeated from one sentence or one clause to another. Sometimes the same idea is expressed again in the next sentence with just a slight change/minor additional information.

Second, there are a lot of places where the felicity of expression is low, making it hard to follow what you want to say (e.g. the passage explaining why you chose the news org at stake).

Third, duplicated words pop up quite often through the manuscript (e.g. line 62-63). 

Fourth, try to cut unnecessary words/phrases line by line. 

Author Response

This paper provides a very interesting case study into the penetration and institutionalisation of data journalism in newsrooms, purportedly through the lens of field theory. The interview data are rich in perspectives and will contribute to the growing body of knowledge about the increasingly central status, roles and functions of data journalism in newswork. The conclusion is sound based on the basis of the empirical data. 

First of all, thank you very much for your comments on my work. I have edited my article taking into account all your criticisms. I hope I have made all the changes you requested in the best way.

That said, I do think the paper still has important work to do:

  1. The most important issue is the lack of a coherence between field theory concepts, the RQs and the data analysis (the Findings section). The Theoretical Framework section introduces the key concepts of field theory but offers very little to link readers to the two RQs espoused earlier.

How do the concepts of cultural, social and symbolic capital and their relationship with habitus help to understand the way data journalism is integrated in news processes and practices (RQ1) and is positioned in newsrooms (RQ2). The Findings section makes this a little more confusing, as it is not organised around the RQs, making it hard to follow your narrative. Of course, I can have a vague idea of how they are linked (esp. after finishing reading the conclusion) but that is not enough. You will need to foreground those relationships explicitly in your literature review and then organise your Findings section accordingly.

One way that helps not only to create a more coherent and focused flow but also improve the theoretical contribution of the paper is to (1) delete the two current RQs from the introduction, then (2), as you go through the theoretical framework, pose a set of new, smaller questions according to field theory's key elements; and then (3) structure your analysis of findings according to each of these smaller questions.

Regarding your suggestion, I made some additions, especially in the literature review part and created a parallel between my research questions and the findings section. I tried to strengthen my findings by referring to other studies in the conclusion section. However, I think I have clearly expressed the connections between my research questions and findings under the headings. Currently, the first three headings in the findings section provide answers to my first research question. The answer to my second research question can be found in the last heading in the findings section.

  1. The Method (not Methodology, but the way) section needs to provide far more details on the interview procedures and processes. Did you, for instance, use a semi-structured interview schedule? If yes, how were the questions ordered to maximise your ability to elicit thoughts and information from your interviewees? Also, one single generic sentence about the data analysis method is not acceptable. Please flesh out more information about the process of thematic coding and how it was used to draw a meaningful path to field theory.

I have added detailed explanations regarding this section in line with your suggestions. I have tried to fill in important gaps in the article, such as how validity and reliability are ensured depending on qualitative research ethics, why and how thematic analysis is done. I have tried to explain how meaningful all these are in field theory with examples from the relevant literature.

  1. The paper needs some major sub-editing and proofreading (see below).

Comments on the Quality of English Language

First, it is too wordy and the pace is at times dragging. There are too many instances in which the same words/phrases are repeated from one sentence or one clause to another. Sometimes the same idea is expressed again in the next sentence with just a slight change/minor additional information.

Second, there are a lot of places where the felicity of expression is low, making it hard to follow what you want to say (e.g. the passage explaining why you chose the news org at stake).

Third, duplicated words pop up quite often through the manuscript (e.g. line 62-63).

Fourth, try to cut unnecessary words/phrases line by line.

I read the paper many times and tried to remove all the mistakes. Now, I think it is a much better paper.

Round 2

Reviewer 1 Report

Comments and Suggestions for Authors

The authors have managed to considerably improve the paper. They have taken into account the suggestions, especially those focused on better focusing the literature review, the description of the methodology and the contrast of the results with other research in the discussion.

Author Response

Firstly, I would like to thank you for all your suggestions and advice. If this study is accepted, it will be thanks to your evaluations of the critical notes.

In the evaluation report, you marked the "can be improved" option for the section titled "Are the research design, questions, hypotheses and methods clearly stated?" Following this, I made a few changes to my article and I believe I have improved it. I revised my research questions with a minor change to make them more understandable. Moreover, by making adjustments in the analysis and conclusion sections, I tried to make my work more understandable and stronger.

Reviewer 2 Report

Comments and Suggestions for Authors

The revised version is a much improved manuscript. It has seriously addressed my previous comments regarding the method section, the quality of writing and, to some extent, the coherence of the entire piece.

I am still not convinced, however, the analysis of findings has been thorough and coherent enough. The author stated in their response that sub-sections 5.1 to 5.3 is the answer to RQ1 on how data journalism is integrated in the newsroom and the last, 5.4., is on how data journalists are positioned in the newsroom. But some of sub-section 5.1 (e.g. the passage between line 332 and line 348) should fall into the answer to RQ2, not RQ1. And if field theory is the guide for this analysis, then most of the entire 5.2. on the forms of capital should be integrated in the answer to the status of data journalism in the newsroom at stake, shouldn't it? 

I have requested a minor revision, but these issues must be considered and addressed before a full acceptance. 

Comments on the Quality of English Language

This is a much cleaner and clearer version. But there are still some minor language errors. Asking for an independent proofreader would help. 

Author Response

Firstly, I would like to thank you for all your suggestions and advice. If this study is accepted, it will be thanks to your evaluations of the critical notes.

After reading your comments, I made some minor adjustments to my research questions to resolve the confusion in my article and made them more relevant to the topic. More importantly, I made some changes in the analysis and conclusion sections of my article. Initially, in the analysis section, I reduced the number of headings from four to three, as you also mentioned, since the third and fourth sections were complementing each other. By merging them into a single heading, I focused on the changing news operations in conjunction with data journalism. Thus, I sought answers to the first research question in the first and second analysis parts and to the second research question in the third part. Furthermore, for clarity and comprehensibility, I made significant changes in the discussion and conclusion sections. I divided the discussion into three main headings and provided concrete and robust recommendations.